# Studying trajectories of multimorbidity: a systematic scoping review of longitudinal approaches and evidence

Genevieve Cezard [1], Calum Thomas McHale [2], Frank Sullivan,[2] Juliana Kuster Filipe Bowles,[3] Katherine Keenan [1]

[1]School of Geography and Sustainable Development, University of St Andrews, St Andrews, UK
[2]School of Medicine, University of St Andrews, St Andrews, UK
[3]School of Computer Science, University of St Andrews, St Andrews, UK

**Correspondence to**
Dr Katherine Keenan;
katherine.keenan@st-andrews.ac.uk

## ABSTRACT

**Objectives** Multimorbidity—the co-occurrence of at least two chronic diseases in an individual—is an important public health challenge in ageing societies. The vast majority of multimorbidity research takes a cross-sectional approach, but longitudinal approaches to understanding multimorbidity are an emerging research area, being encouraged by multiple funders. To support development in this research area, the aim of this study is to scope the methodological approaches and substantive findings of studies that have investigated longitudinal multimorbidity trajectories.

**Design** We conducted a systematic search for relevant studies in four online databases (Medline, Scopus, Web of Science and Embase) in May 2020 using predefined search terms and inclusion and exclusion criteria. The search was complemented by searching reference lists of relevant papers. From the selected studies, we systematically extracted data on study methodology and findings and summarised them in a narrative synthesis.

**Results** We identified 35 studies investigating multimorbidity longitudinally, all published in the last decade, and predominantly in high-income countries from the Global North. Longitudinal approaches employed included constructing change variables, multilevel regression analysis (eg, growth curve modelling), longitudinal group-based methodologies (eg, latent class modelling), analysing disease transitions and visualisation techniques. Commonly identified risk factors for multimorbidity onset and progression were older age, higher socioeconomic and area-level deprivation, overweight and poorer health behaviours.

**Conclusion** The nascent research area employs a diverse range of longitudinal approaches that characterise accumulation and disease combinations and to a lesser extent disease sequencing and progression. Gaps include understanding the long-term, life course determinants of different multimorbidity trajectories, and doing so across diverse populations, including those from low-income and middle-income countries. This can provide a detailed picture of morbidity development, with important implications from a clinical and intervention perspective.

## STRENGTHS AND LIMITATIONS OF THIS STUDY

⇒ This is the first systematic review to focus on studies that take a longitudinal, rather than cross-sectional, approach to multimorbidity.

⇒ Systematic searches of online academic databases were performed using predefined search terms, as well as searching of reference lists, and this is reported using Preferred Reporting Items for Systematic Reviews and Meta-Analyses for scoping reviews guidelines.

⇒ For selected papers, data were double extracted using standardised pro formas to aid narrative synthesis.

⇒ Due to the heterogeneity of the studies included, their weaknesses were described in the narrative synthesis, but we did not perform quality assessment using standardised tools.

within the same individual.[1 2] Multimorbidity represents a huge immediate and future challenge for healthcare systems around the world. It is estimated that 50 million people suffer from multimorbidity in the European Union, and about one in three globally have multiple conditions.[3 4] The global prevalence of multimorbidity is expected to increase through the 21st century, as a result of increased life expectancy, population ageing and the expansion of morbidity. For example, the prevalence of 'complex multimorbidity'—defined as four or more co-occurring chronic conditions—has been projected to increase from about 10% in 2015 to 17% in 2035 in England.[5] The implications of this for individuals and societies are stark: multimorbidity is predictive of poorer quality of life,[6] greater functional decline[7] and increased mortality.[8] Management and treatment of multimorbidity also places a considerable economic and logistical burden on health services,[9] which are not adapted to deal with multimorbidity, being typically organised around the single disease model.

## INTRODUCTION

The term multimorbidity is used to define the co-occurrence of multiple diseases, specifically two or more chronic conditions

In response to this challenge, in the last two decades, there has been an explosion of (predominantly cross-sectional) research that has investigated the risk factors and patterns of multimorbidity. For example, systematic reviews have identified common clusters of diseases,[10–12] which include cardiovascular and metabolic diseases, mental health conditions and musculoskeletal disorders. Common risk factors for multimorbidity include increasing age and low socioeconomic status (SES)[12 13] and poor health behaviours, such as high body mass index and smoking.[14] However, the vast majority of multimorbidity studies apply a cross-sectional approach; longitudinal approaches are scarce. To date, there are more than 70 published systematic reviews about multimorbidity, covering definitions to interventions (eg, refs [2 15]), and none of these focuses on longitudinal studies. While 'snap-shot' analyses are useful for understanding prevalence and clustering of diseases, they provide little information on multimorbidity development over time and sequencing of diseases, which have important implications from a clinical and intervention perspective. Recently, there has been a growing orientation towards longitudinal approaches by academic communities and funders such as the UK's Academy of Medical Sciences.[4]

Therefore, this paper aims to gain an overview of the longitudinal approaches used in multimorbidity research, to better understand what evidence is generated from these approaches and to identify the associated gaps.

Our research questions are:

1. What type and range of longitudinal methods are used to analyse multimorbidity over time within individuals?
2. What are the risk/protective factors identified to be associated with individual multimorbidity trajectories?

We used a scoping review approach to systematically review the emerging body of literature investigating multimorbidity trajectories. Based on a narrative synthesis focused on commonalities and differences, this review provides a methodological summary and a comprehensivereview of the evidence on factors affecting multimorbidity pathways.

## METHODS

We review the literature on longitudinal multimorbidity studies via a scoping review approach rather than using a systematic review or meta-analytic approach.[16] Scoping reviews are adopted when the purpose of the review is to scope a nascent body of literature and appraise gaps.[17 18] In reporting, we follow the recently developed Preferred Reporting Items for Systematic Reviews and Meta-Analyses for scoping reviews (PRISMA-ScR)[19] (online supplemental appendix A).

### Eligibility criteria

Inclusion and exclusion criterion were defined prior to database searches (table 1). A primary eligibility criterion was to measure multimorbidity longitudinally within the same sample of adults using a quantitative approach, and we excluded cross-sectional or qualitative designs, reviews, meta-analyses and commentary that did not contain empirical results. Studies had to measure multimorbidity through recognised diseases/conditions or a defined multimorbidity measure such as the Charlson or Elixhauser comorbidity indices[20 21] but not solely a collection of symptoms/states (such as disability or frailty) or disease risk factors (such as obesity). Studies were required to measure change in multimorbidity between distinguishable diseases rather than progression within a single disease category (eg, different types of cancer). We also excluded studies that examined transitions from an index disease into a secondary disease (eg, comorbidities of diabetes). Finally, included studies were focused on

**Table 1** Study inclusion/exclusion criteria for the scoping review

|  | Inclusion | Exclusion |
|---|---|---|
| Study design | Repeated measures designs, longitudinal quantitative studies, including retrospective and prospective cohort studies. | Cross-sectional studies. Systematic reviews/meta-analyses. Qualitative studies. Expert opinion/committee reports. |
| Methodology | Measure trajectories of multimorbidity longitudinally within the same individuals. Multimorbidity defined as a combination of recognised diseases/conditions (eg, self-report or International Classification of Disease 9th revision (ICD-9) or 10th revision (ICD-10) codes). Trajectories defined as change or accumulation in number of distinguishable diseases. | Different cohorts/samples used across longitudinal study timeline. Multimorbidity defined as combination of symptoms or predisease conditions, that is, not defined ICD-10 diseases (eg, predisease, frailty, disability and quality of life). Transitions or trajectories within a single disease (eg, dementia) or from one disease into another (eg, cancer progression). |
| Population | Adult humans (18+ years). | Infants, children or adolescents (<18 years). Animal research. |
| Publication | Peer-reviewed journal articles. Accessible in English. | Grey literature. Not accessible in English. |

| Table 2 | Summary of search strategy |
|---|---|
| **Search no.** | **Search terms** |
| #1 | Multimorbidity (multimorbid\*; multi-morbid\*) OR Comorbidity (comorbid\*; co-morbidi\*) OR Cooccurrence (cooccur\*; co-occur\*) |
| #2 | disease OR condition OR illness AND cluster\* OR trajectory (trajector\*) OR cascade\* OR accumulation (accumulat\*) OR combination\* OR sequence (sequenc\*) OR transition\* |
| #3 | cohort\* OR longitudinal\* OR prospective\* |
| #4 | #1 AND #2 AND #3 |
| #5 | #4 AND NOT cell\* OR gene OR genes OR bacteria\* OR DNA OR COVID-19 (COVID-19\*) |

adult humans (aged 18+ years) and were peer-reviewed journal articles, written in English language. Our search had no restrictions on date of publication.

## Search strategy

Four online databases were searched: Medline, Scopus, Web of Science and Embase. Initially, scoping searches were conducted within each database, with relevant terms such as 'multimorbidity', 'disease trajectory' and 'longitudinal'. These scoping searches allowed the identification of additional relevant search terms and, where appropriate, Medical Subject Headings (MeSH), in order to develop and refine the final search strategy (table 2).

The final search was a combination of three search elements: first, the concept of multimorbidity, second the methodological approach of disease trajectories and third, longitudinal study design. These search terms initially returned a large number of irrelevant references, focusing on cellular medicine, genetics and COVID-19, so we added an additional condition to exclude these. We also refined the search results to include English language, adult humans and peer-reviewed journal articles only. All searches were conducted in May 2020. The full search syntax is included in online supplemental appendix 1, appendix B. We identified additional relevant papers through recommendations from coauthors and external collaborators. The database search results were searched for these additional papers, and if they were not identified in the database searches, they were included as '*identified through other sources*' and were subject to the same screening procedure as papers identified through database searches.

## Screening and study selection

After deduplication, articles were screened for eligibility by title, abstract and finally full text using Endnote and predefined groups for exclusion reasons and inclusion (work shared between GC, CTM and KK). At abstract and full-text stages, a double screening process was used to minimise evidence selection bias,[22] meaning two coauthors blindly and independently reviewed the study for inclusion. Any disagreements were resolved through discussion and consensus. The reference lists of the selected studies were screened to identify any relevant studies that may have been missed in the main search,

and any newly identified articles were subject to the same screening and data extraction processes.

## Data extraction and synthesis

Three authors (GC, CTM and KK) extracted and double-extracted information on study and sample characteristics, including the title, authors and publication year, study setting, data source used, information on the study population (eg, inclusion and exclusion criteria, sample size and age) and follow-up duration. We also extracted study objectives, multimorbidity conceptualisation and measures, and methodological and analytical approaches, focusing on those specifically used for the analysis of multimorbidity trajectories. Finally, we extracted the key substantive findings and limitations reported in each study in relation to generalisability, accuracy, comprehensiveness, methodology and interpretation.

To develop the narrative synthesis, we analysed and summarised the patterns in the extracted data, investigated the similarities and differences between studies and examined bias and limitations to identify knowledge gaps and the strengths and weaknesses of methodological approaches.

## Ethics approval

This is a review of already published material; therefore, no ethics approval needed.

## Patient and public involvement

No patient involved.

## RESULTS
### Study selection

Figure 1 depicts the study selection process. Database searches returned 11 420 articles and nine additional papers were identified from other sources. Of the combined 11 429 papers, 4705 were duplicate references and removed. Of the remaining 6724 papers, 6315 were removed during title screening and a further 360 papers during abstract screening. The most common reasons for exclusion were studies that did not focus on multimorbidity longitudinally (eg, trajectories were followed within a single disease) and study design not being longitudinal (eg, cross-sectional analysis). The remaining 49 papers went through full-text screening and 19 were

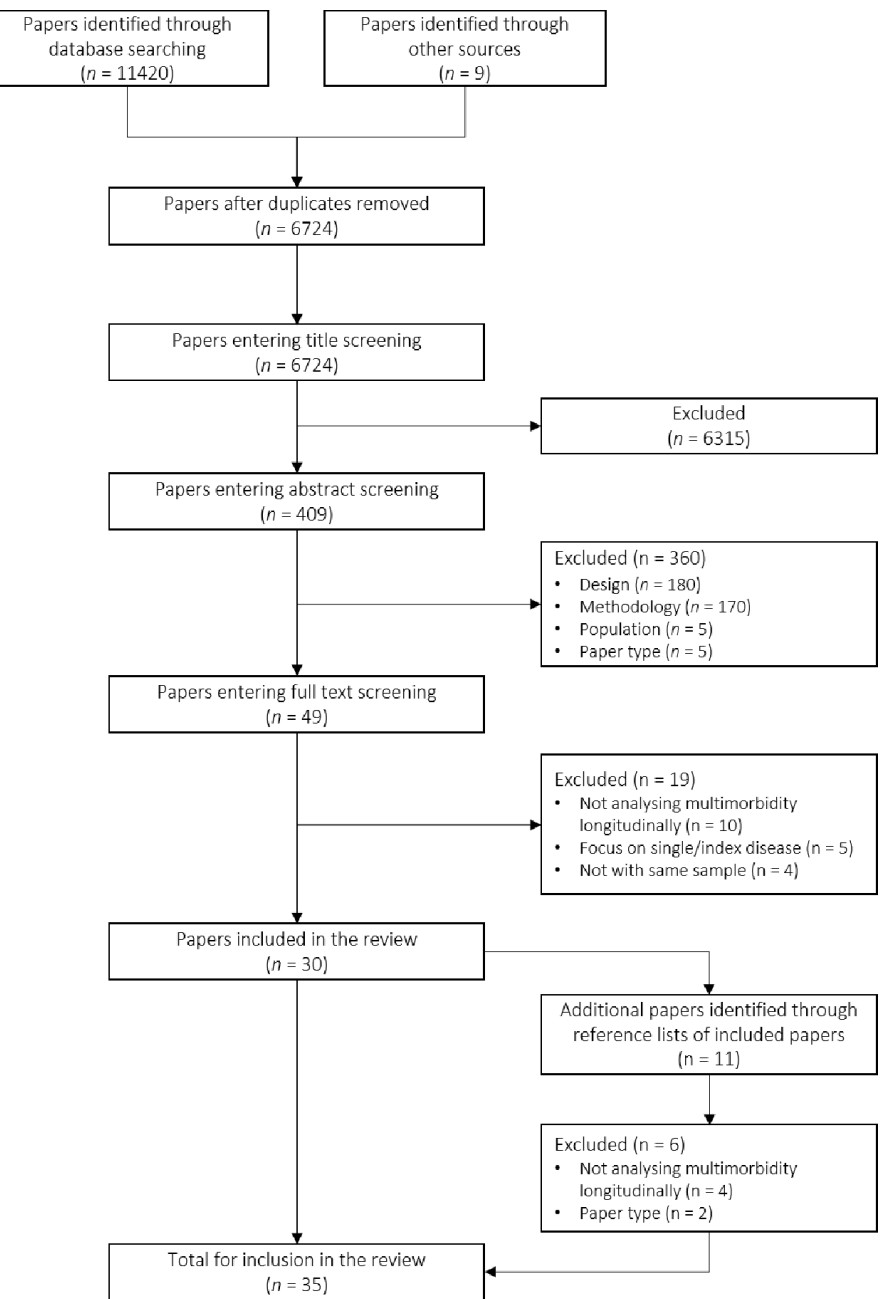

**Figure 1** Study selection process.

subsequently removed. Searching the reference lists of the remaining 30 papers identified another 11 potentially relevant papers. After screening these 11 papers, six were excluded leaving five additional papers for inclusion. In total, 35 papers were selected for further data extraction.

### Characteristics of the selected studies

Table 3 summarises the study characteristics. All articles were published since 2011 and were primarily based on data from European countries (n=16),[23–38] Australia and North America (n=14)[39–52] and high-income Asian countries South Korea, Taiwan and Singapore (n=4).[53–56] Apart from one study using Chinese data,[57] none related to low-income and middle-income settings.

Sample characteristics varied widely, ranging from 756 in a survey of older participants[45] to 6.2 million in a nationwide study using Danish register data,[38] and the length of follow-up periods ranged between 2 and 20 years. Most studies had age restrictions, with about half focused on older populations (50 years+), and one study focused on the very old (80 years plus).[36] Most samples included both males and females apart from two studies including males only[26 37] and three studies including females only.[42 47 51] Three studies focused on US veterans, a predominantly male population.[39 44 48] The data sources used were a combination of administrative data, including primary or secondary care records, disease registries and health insurance data (24 studies), and survey data (18 studies).

**Table 3** Characteristics of the selected studies

| Study, year | Country | Sample size | Baseline sample age | Follow-up period | Data source | Number of conditions included | Disease identification strategy |
|---|---|---|---|---|---|---|---|
| Alaeddini et al, 2017[44] | USA | n=601 805 | Adults | 14 years | Administrative data: inpatient and outpatient data | 4 | International Classification of Diseases, Clinical Modification (ICD-9-CM) codes |
| Ashworth et al, 2019[23] | UK | n=332 353 | 18 years+ | 14.5 years | Administrative data: primary care records | 12 | Read codes in UK primary care (based on the Quality and Outcome Framework) |
| Beck et al, 2016[24] | Denmark | N≈120 000 | All ages | 18.5 years | Administrative data: inpatient and outpatient data, and emergency visits | All diseases taken at three level ICD-10 code | ICD-10 codes (A41 specified for sepsis) |
| Calderón-Larrañaga et al, 2018[31] | Sweden | n=2387 | 60 years+ | 3–6 years | Survey with interviews, clinical examinations, cognitive tests and laboratory tests combined with and administrative data: National Patient Register and death. | 918 diseases grouped into 60 categories | ICD-10 codes |
| Calderón-Larrañaga et al, 2019[32] | Sweden | n=2293 | 60 years+ | 9 years | Survey with interviews, clinical examinations, cognitive tests and laboratory tests combined with and administrative data: National Patient Register and death. | 918 diseases grouped into 60 categories | ICD-10 codes |
| Canizares et al, 2017[40] | Canada | n=10186 | 20–69 years | 18 years | Survey data | 17 | Self-reported |
| Chang et al, 2011[53] | Taiwan | n=147 892 | All ages | 3 years | Administrative data: national insurance claims data including inpatient, outpatient, ambulatory care, dental care and pharmacy expenditure | Not specified | ICD-9-CM codes |
| Dekhtyar et al, 2019[33] | Sweden | n=2589 | 60 years+ | 9 years | Survey with interviews, clinical examinations, cognitive tests and laboratory tests combined with and administrative data: National Patient Register and death. | 918 diseases grouped into 60 categories | ICD-10 codes |

Continued

**Table 3** Continued

| Study, year | Country | Sample size | Baseline sample age | Follow-up period | Data source | Number of conditions included | Disease identification strategy |
|---|---|---|---|---|---|---|---|
| Fabbri et al, 2015[34] | Italy | n=1018 | 60 years+ | 9 years | Survey data with clinical examinations and laboratory tests | 15 | Self-reported medical history, medication use, medical documentation, and clinical examination. |
| Fabbri et al, 2016[45] | USA | n=756 | 65 years+ | 9 years | Survey data with clinical examinations and laboratory tests | 13 | Self-reported, laboratory tests and clinical examination |
| Faruqui et al, 2018[39] | USA | n=257 633 | 18 years+ | 5 years | Administrative data: inpatient and outpatient data | 5 | ICD-9-CM codes |
| Fraccaro et al, (2016)[35] | UK | n=287 459 | 18 years+ | 9.5 years | Administrative data: primary and secondary care, death and out-migration data | 22 | Read codes in UK primary care data (based on validated codes by Khan et al, 2010 [65] |
| Freisling et al, 2020[30] | Seven European countries (Denmark, Germany, Italy, the Netherlands, Spain, Sweden and the UK) | n=291 778 | 35–70 years | 16 years | Administrative data (primary and secondary care, disease registry, health insurance, medication use and death data) and survey (questionnaire at baseline) | 3 | Combination of ICD10 codes and disease registry |
| Gellert et al, 2018[36] | Germany | n=1398 | 80 years+ | 6 years | Administrative data: ambulatory, hospital and long-term care | 30 | ICD-10 codes |
| Hanson et al, 2015[46] | USA | n=41 158 | 65–84 years | 18 years | Administrative data: Utah Population Database, death and Medicare claims | 17 | Diseases are identified following the SEER-Medicare Comorbidity macro from Medicare data in 1992–2009 |
| Hiyoshi et al, 2017[37] | Sweden | n=5218 | 44–51 years | 5 years | Administrative data: cancer register and patient register with inpatient and outpatient data | 18 | Swedish version of ICD-9 and ICD-10 |

Continued

**Table 3** Continued

| Study, year | Country | Sample size | Baseline sample age | Follow-up period | Data source | Number of conditions included | Disease identification strategy |
|---|---|---|---|---|---|---|---|
| Hsu, 2015[54] | Taiwan | n=2584 | 60 years+ | 14 years | Survey data | 15 conditions grouped into six disease types | Self-reported |
| Jackson et al, 2015[47] | Australia | n=4865 | 47–52 years | 12 years | Survey data | 18 | Self-reported |
| Jensen et al, 2014[38] | Denmark | n=6.2 million | All ages | 14.9 years | Administrative data: inpatient, outpatient and emergency visits | All diseases taken at three level ICD-10 code | ICD-10 codes |
| Kim et al, 2018[56] | South Korea | n=121 733 | 60 years+ | 11 years | Administrative data: claims data from private healthcare providers. | 20 | Claims data |
| Lappenschaar et al, 2013[25] | The Netherlands | n=182 396 | 35 years+ | 9 years | Administrative data: primary care data | 6 | International Classification of Primary Care codes, completed with laboratory tests and medication information. |
| Lindhagen et al, 2015[26] | Sweden | n=20237 | Adults | 10 years | Administrative data: cancer and population registers | 17 | ICD codes from patient and cancer registries |
| Perez et al, 2020[27] | Sweden | n=2596 | 60 years+ | 6 years | Survey with interviews, clinical examinations, cognitive tests and laboratory tests combined with and administrative data: National Patient Register and death. | 918 diseases grouped into 60 categories | ICD-10 codes |
| Pugh et al, 2016[48] | USA | n=164 933 (derivation) n=146 051 (validation) | Adults | 3 years | Administrative data: inpatient, outpatient and medications data | Not specified | ICD-9-CM codes |
| Quiñones et al, 2011[50] | USA | n=17 517 | 51 years+ | 11 years | Survey data | 7 | Self-reported |
| Quiñones et al, 2019[49] | USA | n=8331 | 51–55 years | 16 years | Survey data | 7 | Self-reported |
| Rocca et al, 2016[51] | USA | n=3306 | Adults <50 years | 10–19 years | Administrative data: medical records | 18 | ICD-9 codes |

Continued

**Table 3** Continued

| Study, year | Country | Sample size | Baseline sample age | Follow-up period | Data source | Number of conditions included | Disease identification strategy |
|---|---|---|---|---|---|---|---|
| Ruel et al, 2014[52] | Australia | n=1854 | 18 years+ | 6–10 years | Survey (with telephone interview, self-completed questionnaire and clinic biomedical assessment including blood sample) | 8 | Self-reported and biomedical measurements |
| Ruel et al, 2014[57] | China | n=1020 | 20 years+ | 5 years | Survey (with laboratory tests) | 11 | Self-reported, blood samples and clinical examinations |
| Ryan et al, 2018[29] | Ireland | n=4502 | 50 years+ | 2 years | Survey (with health assessment in a health centre) | 30 diseases grouped into 16 categories | Self-reported and clinical examination |
| Siriwardhana et al, 2018[41] | USA (Hawaii) | n=22 930 | 65 years+ | 5 years | Administrative data (health insurance data) | 3 | ICD-9 codes |
| Strauss et al, 2014[28] | UK | n=24 615 (Phase I) n=4532 (Phase II) | 50 years+ | 3 years | Administrative data (primary care records for CiPCA) linked to Questionnaire data (NorStOP) | 42 | Read codes in UK primary care |
| Xu et al, 2018[42] | Australia | n=11 941 | 45–50 years | 20 years | Survey data (self-reported) | 3 | Self-reported |
| Zeng et al, 2014[43] | USA (Colorado) | n=961 (primary cohort) n=13 163 (secondary cohort) | 65 years+ | 10 years | Administrative data (electronic health records including inpatient admission and emergency visits, and death) and survey | 17 | ICD-9-CM codes based on Quan et al, 2005[66] algorithm |
| Zhu et al, 2018[55] | Singapore | N≈700 000 | All ages | 7 years | Administrative data (electronic health records and deaths) | 6 | Not specified |

There were common datasets used across studies, such as the Swedish National study on Ageing and Care in Kungsholmen.[27 31–33] In seven studies, survey data were combined with administrative data sources.[27 28 30–33 43] In five survey-based studies, questionnaire data were supplemented by medical examination records or cognitive or laboratory tests.[29 34 45 52 57] Informed consent of participants was mentioned in 10 of the 18 studies using survey data.

## Methods of disease and multimorbidity ascertainment

Studies based on administrative data relied on clinician-diagnosed diseases, often using standardised diagnosis codes, such as the International Classification of Diseases, that is, International Classification of Disease 9th revision (ICD-9), ICD-9, Clinical Modification (ICD-9-CM), or ICD 10th revision (ICD-10). All survey-based studies used participant self-report for disease identification. Studies that combined survey data with other sources ascertained disease status mostly through clinician diagnosis[27 28 30–33 43] but some supplemented with laboratory and cognitive tests.[29 34 45 52 57] The number of diseases that were considered to contribute to the measure of multimorbidity varied widely, ranging from three[30 41 42] to a very large number based on three levels of ICD-10 codes.[24 38] Studies using survey data used a narrower range of diseases than those drawn from administrative data. The precise list of diseases was never uniform between studies (see appendix C for full details), but the rationale for choosing them was usually described. For example, included diseases with high prevalence and risk of disability and mortality[34 45] or that were assessed/validated by clinicians.[27 28 31–33] Some used lists based on the Charlson and Elixhauser multimorbidity indices,[26 35–37 43 46 56] but these were sometimes augmented with extra conditions[37] or reduced due to data sensitivity restrictions.[35]

## Approaches to the measurement of multimorbidity trajectories

To develop longitudinal measures of multimorbidity, studies tended to take one of two broad approaches. The most common was that repeated measures of multimorbidity status over time were measured for each individual. This mainly involved constructing unweighted or weighted counts of diseases at regular intervals for each individual, thus conceptualising multimorbidity as a continuum (eg, refs [52 57]), although a few still used a binary measure of two or more chronic conditions.[29 40] The second broad approach explored disease transitions.[24–26 30 38 39 41 42 44 55] Only one study explored the order of disease occurrence.[23]

## Types of methodology

Breaking this down further, we identified five broad analytical approaches: constructed variables of multimorbidity change, multilevel regression modelling, transition and data mining methodologies, visual approaches (articles summarised in table 4) and longitudinal group-based methodologies (articles summarised in table 5). Note that some studies employed more than one type of approach (eg, refs [42]). In the first approach, four articles created variables of multimorbidity change.[29 35 53 57] In one study, intraindividual change in Charlson Comorbidity Index (CCI) between baseline and later time points was used,[35] and in another, transitions to two or more conditions or by acquisition of additional conditions.[29] Two studies used simple methods to construct morbidity trajectory groups (eg, 'constant high', 'constant medium' and 'constant low')[53] and disease transition stages (eg, 'healthy' and 'healthy to a single chronic disease').[57] After creating these categorical dependent variables, the authors used them in regression analysis to assess their association with health expenditures,[53] diet[57] and physical activity and functioning.[29]

The next approach, employed by 14 studies (table 4),[27 31–36 40 42 43 45 49 50 52] was multilevel regression modelling (variously referred to as random effects models, growth curve models, hierarchical linear models or multilevel models). These studies analyse repeated measures of multimorbidity within each individual, considering this as a 'trajectory' or 'growth curve'. The dependent variable was typically a count of diseases or a multimorbidity index measured repeatedly, and the coefficients assessed a change in this over time, many including random effects for both the intercept and slope. One study used the regression estimates (ie, intercept and slope coefficients for multimorbidity) to create categories capturing the pace of multimorbidity, for example, 'rapidly accumulating' and 'slowly accumulating', which were used for further modelling.[31] Some of these studies also investigated whether certain covariates such as biomarkers,[27 34] sociodemographics and life experiences[33] affected the pace of change in multimorbidity by including an interaction term between time and the respective covariates.

The next approach, employed by nine studies,[24–26 30 38 39 41 44 55] focused on modelling transitions between specific disease states. Six of the studies focused on a limited number of diseases to make the analysis feasible.[25 30 39 41 44 55] Some studies used principles of state transition modelling, either using Markov principles,[44] acyclic multistate models[41] or state transition modelling.[26 30] Another two studies employed Bayesian techniques including a multilevel temporal Bayesian network[25] and a longest path algorithm to identify the most probable sequence from/to a specific disease following an unsupervised multilevel temporal Bayesian network analysis.[39] One paper derived a disease progression network from real data and used this for further microsimulation.[55] Finally, two studies used a data-driven approach to create 'temporal disease trajectories' by combining significant temporal directed pairs from all disease pairs possible.[24 38] Transition analysis also enabled the identification of longitudinal clusters.[44]

Three studies[23 42 51] used visual methods to describe disease sequences or multimorbidity acquisition sequences. Ashworth et al[23] used alluvial plots to illustrate multimorbidity acquisition sequences based on date

**Table 4** Methods of studies taking four analytical approaches (multimorbidity change variables, regression, transition modelling and visual approaches)

| Approach 1: constructed variables of multimorbidity change (n=4) | | |
|---|---|---|
| **Study, year** | **Main study outcome** | **Statistical methodology used to analyse multimorbidity longitudinally** |
| Chang et al, 2011[53] | Prospective medical use and expenditures | Morbidity trajectory groups: constant high, constant medium, constant low, decreasing, increasing and erratic. |
| Fraccaro et al, 2016[35] | Time to death from any cause | Change in comorbidities: difference between baseline Charlson Comorbidity Index (CCI) and 1-year, 5-year and 10-year follow-up CCI scores as a proportion of mortality rate. |
| Ruel et al, 2014[57] | Six chronic disease transition stages groups | Groups of chronic disease transition stages (healthy, healthy to a single chronic disease, stable with a single chronic disease, healthy to multimorbidity, stable multimorbidity and increasing multimorbidity) |
| Ryan et al, 2018[29] | Development/worsening of multimorbidity | Constructed measures of multimorbidity change (from zero or one condition at baseline to two or more conditions) and of worsening of multimorbidity (from multimorbidity at baseline to additional conditions at follow-up) |
| **Approach 2: regression-based approaches (n=14)** | | |
| Calderón-Larrañaga et al, 2018[31] | Disability (IADL* count) | Linear mixed models† estimated the rate of multimorbidity accumulation. Predicted slopes are examined by quartile and further dichotomised into 'rapidly accumulating' (upper quartile) and 'slowly accumulating' (three lower quartile). |
| Caldeón-Larrañaga et al, 2019[32] | Level of multimorbidity (count of chronic conditions) and disability (IADL count) | Linear mixed models assessed the association between baseline level of psychological factors and multimorbidity and disability over time. The interaction terms between time and each of the psychological factors were included as a fixed effect. |
| Canizares et al, 2017[40] | Multimorbidity (binary – two or more conditions) | Multilevel logistic growth modelling was used to examine the age, period and cohort effects on multimorbidity. Observations nested in individuals and age and birth cohort entered as fixed effects. |
| Dekhtyar et al, 2019[33] | Level of multimorbidity (count of chronic conditions) | Linear mixed models assessed the association between life experiences and the speed of multimorbidity accumulation. Interaction terms between time and life experiences included as fixed effects. |
| Fabbri et al, 2015[34] | Level of multimorbidity (count of chronic conditions) | Linear mixed models assessed the association between baseline age, disease status and biomarkers with the number of diseases over follow-up. The study also tested whether increased Interleukin 6 (IL-6) over time would predict steeper increase in multimorbidity over time, independent of baseline IL-6. |
| Fabbri et al, 2016[45] | Standardised neurocognitive tests evaluating cognitive function | Linear mixed models were used to estimate rate of change in multimorbidity (count of diseases). The individual slopes of multimorbidity rise were dichotomised into faster accumulation (upper quartile) and the rest (lower three quartiles). |
| Fraccaro et al, 2016[35] | All-cause mortality | Multimorbidity change was measured by differences between baseline CCI and 1-year, 5-year and 10-year follow-up CCI scores as a proportion of mortality rate. Survival analysis (Cox regression) estimated mortality rates as a function of age, gender and CCI scores (fixed and time varying). |
| Gellert et al, 2018[36] | Level of multimorbidity (count of comorbidities-based on Elixhauser) | Linear mixed models (random intercepts and slopes) estimated differential increase in the number of comorbidities over 25 calendar quarters prior to death in centenarian, nonagenarian and octogenarian cohorts. |
| Perez et al, 2020[27] | Level of multimorbidity (count of chronic conditions) | Linear mixed models (random intercepts and slopes) were employed to analyse the association between baseline total serum glutathione levels and level of multimorbidity. |
| Quñones et al, 2011[50] | Level of multimorbidity (count of chronic conditions) | Linear mixed models (random intercepts and slopes) analysed ethnic variations in level of multimorbidity. |
| Quñones et al, 2019[49] | Level of multimorbidity (count of chronic conditions) | Negative binomial generalised estimating equation (GEE) models with a first-order autoregressive covariance structure were used to assess the relationship between chronic disease accumulation and race/ethnicity. |
| Ruel et al, 2014[52] | Count of chronic conditions+incidence of multimorbidity | Multinomial logistic regression estimated the count and individual proportion of chronic diseases in those with no or one chronic disease at baseline. |
| Xu et al, 2018[42] | Cumulative incidence of three conditions (diabetes, heart disease and stroke) | Repeated measures logistic regression using GEEs were used to identify risk factors for developing three conditions and their combinations. Generalised linear mixed models were used to estimate the associations between predictors and the progression to multimorbidity. |

Continued

**Table 4**   Continued

| Approach 1: constructed variables of multimorbidity change (n=4) | | |
| --- | --- | --- |
| **Study, year** | **Main study outcome** | **Statistical methodology used to analyse multimorbidity longitudinally** |
| Zeng et al, 2014[43] | Self-reported health, number of primary care visits, inpatient admissions, emergency department visits and mortality | Linear mixed models (random intercepts and slopes) estimated the individual trajectory of CCI over time (up to 10 years), which was used as an independent variable in a subsequent linear regression model for the health outcomes. |
| **Approach 3: transition and disease progression modelling (n=9)** | | |
| Alaeddini et al, 2017[44] | Clusters of disease transition considering four conditions (hypertension, depression, post-traumatic stress disorder and back pain) | Disease transitions were modelled using Markov chain models with a transition matrix, placed in a Latent Regression Markov Mixture Model to incorporate subject–specific covariates (eg, age, sex, race/ethnicity, etc). Markov Clustering algorithm was used to identify patterns of disease progression. |
| Beck et al, 2016[24] | 30-day mortality in patients with sepsis | Data-driven method combining temporal directed pairs for identification of disease trajectories based on the method developed by Jensen et al[38] |
| Freisling et al, 2020[30] | Transition to cancer-cardiometabolic multimorbidity | Non-Markovian multistate modelling for transitions to cancer, cardiovascular disease (CVD), type 2 diabetes and subsequently to multimorbidty using cox proportional hazards. |
| Faruqui et al, 2018[39] | Development of five specific conditions | Comparison of several methods: unsupervised Bayesian network, multivariate regression and latent regression Markov mixture modelling. Longest Path Algorithm from the Bayesian network was used to identify the most probable sequence from/to a specific disease. |
| Jensen et al, 2011[38] | Disease trajectories | Temporal correlation analysis, that is, strength of correlation between the pair of diseases (relative risk >1) for over a million pairs where disease 2 (D2) occurs within 5 years of disease 1 (D1) and based on directionality (whether D1->D2 occurs more often than D2->D1, binomial tests). Disease trajectories combining pairs with overlapping diagnosis, into three or more diseases. |
| Lindhagen et al, 2015[26] | Mortality; change and scale of CCI change | A state transition model in discrete time steps to estimate changes in CCI. Transition probabilities were estimated using logistic/Poisson regression models for vital status and CCI changes. Simulation models estimated changes in CCI with their CIs. |
| Lappenschaar et al, 2013[25] | Cumulative incidence and combinations of six cardiovascular diseases | Multilevel temporal Bayesian networks were used to model the patient's disease status at baseline and 3–5 years after. The variance induced by the urbanisation level, age and gender in the multilevel model was explained using Markov Chain Monte Carlo simulation. |
| Siriwardhana et al, 2018[41] | Disease state probability and transition probability from a single disease state to a multiple disease state. | Acyclic multistate model to define an interconnected progressive chronic disease system for the elderly population. Aalen and Johansen estimator (a non-parametric technique) to estimate marginal state occupational probabilities. |
| Zhu et al, 2018[55] | Disease progression states and the absorbing state, death. Life years lost to a specific condition and cumulative lifetime risk of certain conditions | A disease progression network was constructed based on the real cohort. One-year progression from state A to B is calculated by counting the number of people who are in state A the previous year and in state B the following year. Microsimulation is used to calculate life years lost and lifetime risk of particular states. |
| **Approach 4: visualisation methods (n=3)** | | |
| Ashworth et al, 2019[23] | Level of multimorbidity (count of chronic conditions) | Alluvial plots based on date of onset of each long-term condition, tabulated as first, second and third to visualise the acquisition sequence. |
| Rocca et al, 2016[51] | Level of multimorbidity (count of chronic conditions) | The accumulation of multimorbidity was represented graphically using Aalen-Johansen curves (a multistate generalisation of cumulative incidence curves; unadjusted curves considering all 18 conditions equally). |
| Xu et al, 2018[42] | Cumulative incidence of three conditions | Sankey diagram was constructed to characterise the dynamic changes of different combinations of the three conditions over time. |

*IADL=limitations in instrumental activities of daily living.
†Linear mixed models here refer to any multilevel model for repeated measures over time for each individual, incorporating various labels – mixed linear model, hierarchical models, growth curve model, etc.

**Table 5** Methods of studies investigating multimorbidity trajectory groups

| Study, year | Statistical approach used | Trajectory groups identified |
|---|---|---|
| Hanson *et al*, 2015[46] | Finite mixture modelling (Proc TRAJ in SAS) was used, with a zero-inflated distribution. Optimal number of groups determined by Bayesian Information Criterion (BIC). | Six groups: 'robust' (no conditions), 'initiates' (none at baseline, increase over time), 'slow initiates' (some at baseline and gradual increase over time), 'accelerated initiates' (none at baseline and quick increase followed by deceleration), 'chronic low' (steady comorbidity over time), 'ailing' (moderate levels of comorbidity at baseline and steady increase over time), 'frail' (high comorbidity at baseline, remaining high over time). |
| Hiyoshi *et al*, 2017[37] | Group-based trajectory modelling, using a zero-inflated distribution. Optimal number of groups determined using BIC. | Four groups identified: 'a constant low trajectory', 'a low start and an acute increase trajectory', 'medium start and a slow increase trajectory' and 'a high start and a slow increase trajectory'. |
| Hsu, 2015[54] | Multiple group-based trajectory model (Proc TRAJ in SAS). Morbidity was set to follow a logistic model. The optimal group number was determined using the BIC and parsimony principle. | Four chronic disease trajectories were identified: 'low risk', 'cardiovascular risk only', 'gastrointestinal and chronic non-specific lung disease' and 'multiple risks'. |
| Jackson *et al*, 2015[47] | Latent class growth analysis (LCGA) in Mplus; optimal group number determined using BIC. | Five groups identified: 'no morbidity, constant', 'low morbidity, constant', 'moderate morbidity, constant', 'no morbidity, increasing' and 'low morbidity, increasing'. |
| Kim *et al*, 2018[56] | Growth mixture modelling in SAS was used; optimal group number determined using BIC. | Five groups identified: 'consistently low', 'increased', 'decreased (low)', 'decreased (high)' and 'consistently high'. |
| Pugh *et al*, 2016[48] | Latent Class Analysis, based on the distribution of repeated measures in the 20 binary diagnosis outcomes. | Five groups identified for both men and women: 'Healthy', 'Chronic Disease', 'Mental Health', 'Pain' and 'Polytrauma Clinical Triad (PCT pain, mental health and traumatic brain injury'. Two additional classes found in men were 'Minor Chronic' and 'PCT with chronic disease'. |
| Strauss *et al*, 2014[28] | LCGA in Mplus, optimal group number determined by iterative modelling and BIC values and likelihood ratio test. Morbidity counts were assumed to be Poisson distributed. Quadratic growth curves were applied for all groups identified within the LCGA models. | Five groups identified and validated: 'no recorded chronic problems', 'developed a first chronic morbidity over 3 years', 'a developing multimorbidity group', 'increasing number of chronic morbidities' and 'a multichronic group with many chronic morbidities'. |

of disease onset, and although useful to understand the order of diseases (co-)occurrence, the visualisations are unable to account for the pace of multimorbidity progression. Aalen-Johansen curves (a multistate generalisation of cumulative incidence curves) were used to represent the accumulation of multimorbidity graphically[51] and the Sankey diagram to show the longitudinal progression and transitions to each disease and disease combinations.[42]

The final approach, employed by seven studies,[28 37 46–48 54 56] was to construct meaningful categories of longitudinal multimorbidity patterns (and associate these with other covariates (summarised in table 5). Methodologies included latent class analysis, latent class growth analysis, growth mixture modelling or group-based trajectory modelling and typically identified between four and six groups of distinct longitudinal multimorbidity patterns. Two studies took an associative approach to explore that specific diseases cluster longitudinally.[48 54] For example, Hsu[54] found four trajectory groups: 'low risk', 'cardiovascular risk only', 'gastrointestinal and chronic non-specific lung disease' and 'multiple

risks'. The other five studies focused on stages of accumulation. Hiyoshi *et al*[37] found four trajectory groups ranging from 'a constant low trajectory' to 'a high start and a slow increase trajectory'. Generally, these clusters incorporated data on the initial level of multimorbidity, and accumulation pattern over time, and nearly all showed accumulation (the exception being Kim *et al*,[56] which identified some groups with decreasing morbidities).

### Results of the studies: outcomes and risk factors
#### Prediction of other health outcomes
Seven of the studies used multimorbidity trajectories to predict subsequent health outcomes,[31 35 43 45 53 55 56] including self-reported health, cognitive ability, disability, medical utilisation and mortality (table 6). Among older adults, results showed that an increase in multimorbidity over 10 years was associated with worse reported health[43] and that those who developed multimorbidity faster had greater risk of disability.[31] In one study, changes in multimorbidity were found to be more predictive of mortality than baseline multimorbidity.[35] By contrast, another

**Table 6** Summary of association analysis for health outcomes related to longitudinal multimorbidity trajectories

| Study, year | Outcome investigated | Findings of association analysis |
|---|---|---|
| Calderón-Larrañaga et al, 2018[31] | Disability | The speed of multimorbidity is a strong predictor for disability in older adults, even when accounting for baseline number of chronic conditions. |
| Chang et al, 2011[53] | Medical utilisation | Morbidity strata predicted medical utilisation as usefully as more complex risk adjusters. |
| Fraccaro et al, 2016[35] | Mortality | Change over time of Charlson Comorbidity Index (CCI) was a stronger predictor of mortality than baseline CCI. |
| Fabbri et al, 2016[45] | Standardised neurocognitive tests | Accumulation of multimorbidity was associated with faster decline in verbal fluency but seems to have no effect on memory decline, in older adults without mild cognitive impairment or dementia. |
| Kim et al, 2018[56] | Mortality | The 'consistently high' multimorbidity trajectory group had the highest risk of mortality at 1-year, 3-year and 5-year follow-ups. |
| Zeng et al, 2014[43] | Self-reported health, number of primary care visits, inpatient and emergency admissions and mortality | Growth curve models gave marginally better fitting models for the outcomes of self-reported general health status, but mortality and inpatient status was best predicted by multimorbidity snapshot prevalence the year before the survey. |
| Zhu et al, 2018[55] | Life expectancy | Diabetes, plus hypertension plus complications reduced life expectancy the most. The earlier the onset of multimorbidity, the greater the reduction in life. |

study confirmed that a change in CCI predicts mortality but not necessarily better than a cross-sectional estimate of multimorbidity.[43] Zhu et al[55] found that earlier development of chronic conditions and earlier complications incur greater life-years lost. Finally, multimorbidity accumulation (as a marker of physical health deterioration) predicted faster decline in verbal fluency in older adults without cognitive impairment or dementia.[45]

### Risk factors for multimorbidity

Nineteen of the selected articles[23 25 27–30 32–34 37 40–42 46 47 49 50 54 57] investigated risk factors for multimorbidity trajectories (table 7). Increasing age, although often accounted for in analyses, emerged as a dominant risk factor for acquisition, worsening or progression of multimorbidity.[29 42] As expected, younger age groups were more likely to belong to a non-chronic healthier cluster.[28] However, trajectories starting with depression were more prevalent in younger individuals.[23] Younger cohorts were also found to be more likely to develop multimorbidity and to do so at a younger age.[40] A few studies reported gender differences, with conflicting results. While one study found that those in the 'multiple risks' group were more likely to be female,[54] another two studies found that men were more likely to transition between disease states than women.[30 41]

Four studies investigated ethnic variations.[23 41 49 50] In two US studies, compared with non-Hispanic whites, black Americans had a higher rate of multimorbidity at baseline along with a slower rate of disease accumulation over time, while Hispanic participants tended to start with fewer diseases and increase more rapidly.[49 50] Different ethnicities also had different disease transition patterns. In the USA, white individuals were more likely to transition from ischaemic heart disease to death, while

Asian and Native Hawaiian and Pacific Islander individuals were more likely to transition from diabetes to diabetes plus chronic kidney disease.[41] In the UK, disease-specific sequences also differed by ethnicity: for example, the white ethnic group was dominated by depression as a starting point, while diabetes was the most common starting point in the black ethnic group.[23]

The studies also explored a range of sociodemographic determinants including area-based deprivation, education, occupation, income and marital status. Results largely confirm those found with cross-sectional analyses, with lower SES associated with worse multimorbidity trajectories. For example, lower levels of education were associated with higher rate of multimorbidity accumulation[33 42] or worse multimorbidity trajectories.[47] People living in more deprived areas were more likely to be in an evolving or multichronic multimorbidity cluster[28] and to have trajectories with diabetes and depression as the most common starting point.[23]

Health and health behaviours also showed associations. A Chinese study showed that a greater consumption of fruits, vegetables and grain slowed the development of multimorbidity.[57] Alcohol consumption, smoking and physical inactivity were associated with worse multimorbidity trajectory patterns.[30 42 47] Physical function (measured by gait speed and grip strength at baseline) was associated with development and worsening of multimorbidity over 2 years in a sample of adults aged 50 years and over.[29] Being overweight or obese was also associated with developing or worsening multimorbidity trajectory.[29 42 47] Two studies investigated the role of specific biomarkers, finding that chronic inflammation, system dysregulation and multisystem failure are associated with

**Table 7** Summary of association analysis for risk factors related to multimorbidity trajectories

| Study, year | Risk factors | Findings of association analysis |
|---|---|---|
| Ashworth et al, 2019[23] | Age, ethnicity and deprivation | Trajectories varied by age, ethnicity and deprivation. Depression as a starting point was more common in younger, more deprived and white ethnic group. |
| Caldeón-Larrañaga et al, 2019[32] | Attitude towards life and health | Better attitudes towards life and health were associated with slower multimorbidity development, independent of demographic, clinical, social, personality and lifestyle factors. |
| Canizares et al, 2017[40] | Birth cohort | In each succeeding cohort, multimorbidity rates was higher and multimorbidity emerged earlier. Differences persisted independently of the risk factors for multimorbidity and period effect. |
| Dekhtyar et al, 2019[33] | Elementary education (early adulthood), lifelong active occupation (mid-adulthood), social network (later life) | Adults over 60 years old with higher than elementary education, lifelong active occupations and richer social networks had slower multimorbidity accumulation. The association between childhood circumstances and multimorbidity accumulation was attenuated by subsequent (mid and late) life experiences. Rich social networks reduced the speed of disease accumulation irrespective of lifelong job stress and level of education. |
| Fabbri et al, 2015[34] | Biomarkers: IL-6, IL-1ra, TNF-$\alpha$ receptor II, and DHEAS (as a marker of chronic inflammation and system dysregulation) | Multimorbidity development with age was not linear, and significantly accelerated at older ages. Higher IL-6, IL-1ra and TNF-$\alpha$ receptor II and low DHEAS were associated with higher multimorbidity at baseline, independent of age, sex, BMI and education. Higher IL-6 and steeper increase in IL-6 predicted an accelerated rise in multimorbidity over 9 years of follow-up. |
| Freisling et al, 2020[30] | Sex, age, healthy lifestyle (healthy lifestyle index): diet (Mediterranean Diet Score), alcohol, smoking status and duration, physical activity (Cambridge index) and BMI. Education, menopausal status, use of hormones for postmenopausal women. | Healthy lifestyle habits were strongly associated with lower incident multimorbidity of cancer and cardiometabolic diseases. The risk of transitioning to multimorbidity after having developed a first of the three chronic diseases was higher in men than in women. |
| Hanson et al, 2015[46] | Parity, timing of childbearing, birth outcomes of offspring | High parity, early childbearing and adverse offspring birth outcomes are associated with particular later-life comorbidity patterns and trajectories, when controlling for early-life conditions (age at parental death, childhood socioeconomic status, familial excess longevity and religious participation). |
| Hiyoshi et al, 2017[37] | Income and marital status | Income and physical, cognitive and psychological function were associated with trajectory group membership in unadjusted analysis but not in fully adjusted analysis. |
| Hsu, 2015[54] | Gender, education, physical function, depressive symptoms, life satisfaction, number of health examination, smoking and drinking. | Those in the 'multiple risks' group were more likely to be female, less educated, with more physical function difficulties, more depressive symptoms, lower life satisfaction, more health examinations and not to smoke or drink. Members in the 'CVD risk only' and 'multiple risks' groups were more likely to have physical function difficulties and depressive symptoms. |
| Jackson et al, 2015[47] | Overweight or obesity, education, difficulty managing income, smoking alcohol consumption and physical activity | Being overweight or obese, having a lower education level and difficulty managing on income associated with belonging to an accumulation trajectory. Smoking, alcohol intake and physical activity level also appeared to be important risk factors for the development of some trajectories. |

Continued

**Table 7** Continued

| Study, year | Risk factors | Findings of association analysis |
|---|---|---|
| Lappenschaar et al, 2013[25] | Urbanisation, multimorbidity at baseline | Urbanisation level of a general practice is associated with the higher cumulative incidence of chronic cardiovascular conditions, in particular obesity, hypertension, dyslipidaemia, diabetes mellitus and ischaemic heart disease. Disease accumulation rate higher when multimorbidity is already present at baseline. |
| Perez et al, 2020[27] | Total serum glutathione (biomarker of multisystem failure) | Lower baseline levels of total serum glutathione were associated with a higher rate of multimorbidity development, independent of covariates. |
| Quñones et al, 2011[50] | Race/ethnicity | White Americans differ from black and Mexican Americans in terms of level and rate of change of multimorbidity. Mexican Americans demonstrate lower initial levels and slower accumulation of comorbidities relative to white American. In contrast, black Americans showed an elevated level of multimorbidity throughout the 11-year period of observation, although their rate of change slowed relative to white Americans. |
| Quinones et al, 2019[49] | Race/ethnicity | Non-Hispanic black respondents had higher initial chronic disease counts, but slower accumulation rates, than non-Hispanic white respondents. Hispanic respondents had lower initial chronic disease counts but faster accumulation than non-Hispanic white respondents. |
| Ruel et al, 2014[57] | Dietary patterns | Greater amount of fruits and vegetables and grain (other than rice and wheat) associated with reduced accumulation of multimorbidity. |
| Ryan et al, 2018[29] | Multimorbidity at baseline, age, obesity, gait speed and grip strength and access to government funded primary care. | In non-multimorbid participants age, obesity, gait speed and grip strength were significantly associated with development of multimorbidity. Age, access to government funded primary care, gait speed and grip strength were significantly associated with worsening of multimorbidity in those with multimorbidity. Gait speed and age were significantly associated with new condition development in people with complex multimorbidity. (Overall) Gait speed, grip strength and age were significantly associated with both the development of multimorbidity and accrual of additional conditions with evidence of a dose-–esponse relationship. |
| Siriwardhana et al, 2018[41] | Age, sex and race/ethnicity | Men were more likely to transition between states than women. Whites had the highest risk of transitioning from ischaemic heart disease to death. Asians and Native Hawaiian and Pacific Islanders were more likely to transition from diabetes to diabetes and chronic kidney disease. |
| Strauss et al, 2014[28] | Age and deprivation | Younger age groups were more likely to be in the non-chronic cluster than older groups. Females were more likely to develop or start with multimorbidity than males. More deprived individuals were more likely to be in the evolving (rather than static) multimorbidity cluster. |
| Xu et al, 2018[42] | Sex, age, marital status, income, education, obesity, physical activity, smoking and immigrant status. | Odds of multimorbidity progression increased over time and with age. Women with stroke were more likely to progress to another disease and become multimorbid than other baseline characteristics. In adjusted models, accumulation of multimorbidity was associated with non-married status, low income, lower education, obesity, sedentary and smoking, and immigrant status. Obesity differently associated with different sequences. |

BMI, body mass index; CVD, cardiovascular disease; IL-6, interleukin 6; TNF-α, tumour necrosis factor.

faster rate of multimorbidity accumulation.[27 34] There were associations with family factors: being married was found to be protective of greater multimorbidity accumulation,[37 54] and young parenthood (younger than 25 years) and extremely high parity (nine of more births) significant risk factors.[46] Finally, a negative attitude towards life and health such as low life satisfaction and negative health outlook was associated with poorer multimorbidity trajectories.[32]

## DISCUSSION

Understanding longitudinal multimorbidity trajectories is an important public health priority for clinicians, academics and funders alike.[4] This review aimed to take a systematic approach to scope existing research in the field with a focus on summarising commonly used methodological approaches and substantive findings. In doing so, we provide, to our knowledge, the first review to address longitudinal studies of multimorbidity, in a field

saturated by cross-sectional research.[2 12 15 58] A strength of this review is the systematic and robust approach taken to searching and screening articles for inclusion and reviewing the selected studies, which should limit selection and extraction bias. We used predefined search terms, inclusion criteria and data extraction tools, and we engaged in double screening and extraction.[22] The scoping review process meant that we summarised a wide variety of evidence, and therefore, it was not possible to perform a meta-analysis or use a standardised critical appraisal tool. Nevertheless, we provide a narrative-style critical summary of the selected articles. The results demonstrate that despite widespread expressed interest, relatively few studies do take a longitudinal approach to multimorbidity. All the studies included were published within the last decade and the vast majority using data from high-income countries. The studies showed a great variability in sampling strategy, ways of measuring multimorbidity and statistical approaches to characterising multimorbidity longitudinally. Methods for identifying longitudinal patterns ranged from counts of diseases to cluster or group-based analyses, to modelling transitions between diseases or disease sequences, and these were differentially useful for modelling accumulation, sequencing, clusters or transitions. From a substantive perspective, the studies showed associations with adverse outcomes such as worse reported health, greater risk of disability and mortality that we might expect based on the existing cross-sectional research. A range of multimorbidity trajectory risk factors were also identified, including sociodemographic factors, health behaviours, physical function, biomarkers, marriage and fertility factors, and attitudinal factors.

A limitation of narrative reviews is that they might select evidence to support a particular stance and do not necessarily take enough steps to eliminate selection bias. However, we selected a comprehensive set of items to extract before starting the review, and we engaged in double screening and extraction. Therefore, our methodological approach should limit selection and extraction bias. Our review did not engage in a critical appraisal of the quality of the selected studies. However, when the aim of a scoping review is to provide an overview of evidence (as ours was), methodological limitations and risk of bias of the evidence are not necessarily relevant and generally not performed.[18]

The review has highlighted some geographical bias in the distribution of multimorbidity research. In particular, there was an under-representation of longitudinal multimorbidity research in low-income and middle-income countries (LMICs), which likely reflects the geographical focus of multimorbidity research more generally.[59] This may be due to underinvestment in multimorbidity research in LMICs, coupled with challenges of collecting or accessing relevant data. For example, most of the selected studies used electronic medical records or large-scale longitudinal surveys, which are rare in developing countries. Nevertheless, due to the population ageing trends in LMICs, multimorbidity is already a major public health issue, with potentially more complex comorbidity patterns (eg, undiagnosed conditions or interactions with infectious disease), which deserve research using a longitudinal approach. Recently, published work in LMICs countries has tended to employ a cross-sectional design to analyse multimorbidity[60–63] and therefore were not eligible for inclusion in this review. In addition, none of the studies in the review made cross-country comparisons, which may help to generate stronger evidence about disease trajectories and mechanisms involved in multimorbidity development and progression. For example, comparable cross-country patterns may suggest common biological mechanisms, whereas divergent findings could suggest moderation or prevention of disease processes by policy approaches to treatment, healthcare settings and institutional structures.

The selected studies used a great variety of data sources including administrative data (primary and secondary care data, health insurance claim data, patient and disease registries) and survey data, leading to variations in sample size and issues of generalisability. Issues of small sample size were only discussed in a limited number of studies, mostly in relation to subgroups such as ethnic minorities.[23 41 49 50] Despite the use of large surveys or administrative data, the majority of studies expressed doubts about the generalisability of their findings. For example, well-educated and wealthy individuals were reported as over-represented in longitudinal survey samples.[27 29 31–33 42 47] Studies using administrative data sources typically investigated multimorbidity based on complete follow-up and excluding those who died, generating immortal time bias and investigating potentially healthier populations.[39 53] In other studies, the choice of data sources themselves induced bias, for example, where samples were based on health service users.[35 48] Others explained their sample might be representative but only of a particular group in a specific region (eg, Utah[46]). Another issue of generalisability, mentioned in previous reviews,[15] was related to the heterogenous multimorbidity measures used.[64] A wide variety of different diseases were included, and only a few studies used 'standard' measurement of multimorbidity like the Charlson[20] or Elixhauser[21] indices. Due to the diversity of data sources, diseases were ascertained in multiple ways, using clinical diagnosis, laboratory results, medication use and self-report. The only common measurement feature was that studies in this review tended not to define multimorbidity as the presence or two or more diseases.

The choice of statistical methods served to highlight or obscure different aspects of multimorbidity. For example, the most common approach, multilevel or single-level regression modelling, emphasises accumulation, providing the opportunity to simultaneously evaluate the baseline level of multimorbidity and the (slope) change in multimorbidity and how this differs between groups with different characteristics. However, it tends to obscure the role of specific diseases by collapsing all

morbidity in a single count or index, and we cannot tell, for example, whether this faster accumulation is predominantly occurring among certain types of disorders. Complementary to regression approaches, grouped-based methodologies aimed to classify individuals into types of multimorbidity accumulation. A minority of studies employed the cluster-based approach to understand how specific diseases co-occur over time,[48 54] which extends cross-sectional approaches often referred to as associative multimorbidity.[11] This has the advantage of providing a more detailed understanding of the constellation of diseases that contribute to distinct trajectories, but due to the rarity of some diseases, will tend to find only highly prevalent clusters and is not suitable for rarer disease trajectories.

Some studies conceptualised longitudinal multimorbidity as transitioning between different disease states, using either structured Markov frameworks,[44] multistate modelling[26 41] or a more data-driven, unsupervised approaches.[24 38] The former, more structured approach to disease transition tended to provide a very detailed understanding of interactions between a small set of diseases, which can provide useful evidence for targeting prevention at those with the first disease, a risk stratification approach. The latter, data-driven approaches provide very comprehensive evidence for population-based strategies but relies on large datasets collected over a number of years and appropriate clinical expertise to interpret the results of patterns identified through artificial intelligence (AI). Given the growing interdisciplinary collaborations between epidemiology and computer science, data-driven research will continue to expand in the coming years and extend to prediction modelling and projections. One of the strengths of computer science, and the recent new developments in AI with machine learning, is the ability to work towards solutions that can combine prediction models and compare different treatment options for cohorts of patients (eg, what is the likelihood that a medication commonly used for one chronic condition may speed up the progression of another condition or lead to the development of a new condition).

Compared with cross-sectional studies, longitudinal approaches provide more detailed insight about the role of specific risk factors. For example, while age is a known risk factor, this review highlights how older individuals, once multimorbid, show acceleration of multimorbidity.[29] Multimorbidity trajectory patterns varied by ethnicity,[23 41 49 50] marital status,[37 42] educational level and area-level deprivation,[28 33 42 47] confirming some patterns observed in cross-sectional data. A useful exploitation of longitudinal data–not included in these studies—would be to explore how change in risk factors such as SES or marital status influences different multimorbidity trajectories, which may help identify at-risk groups and target prevention strategies. As highlighted by Zhu and colleagues,[55] the earlier the multimorbidity onset in the life course, the greater the life year lost for that individual. Therefore, future research should seek to take a life course approach in order to disentangle early preventable factors of multimorbidity onset but also to determine later life factors influencing additional disease accumulation. Risk factors should be considered at the level of the individual (life course and contemporaneous factors), medication use and the wider social environment, including poor environmental conditions, and interaction with institutional structures (eg, healthcare system organisation). The increasing availability of 'big data', which links longitudinal administrative data on individuals with health, and geospatial data will make these holistic approaches technically possible. Future research should focus on generating the knowledge required to develop interventions aimed at preventing both the onset and the worsening of multimorbidity.

## CONCLUSION

This review identifies a small but developing body of literature attempting to describe multimorbidity longitudinally. There was a notable lack of studies in LMICs, as well exploring minority ethnic groups. A wide variety of complementary methods are employed, emphasising factors associated with greater disease accumulation, speed of accumulation and specific disease transition processes. Methodologies based on disease ordering or sequence was seldom explored by the studies, and while it is challenging to identify exact timing of disease, future research could seek to investigate disease sequencing that underlies the accumulation process. Risk factors for trajectory types could inform future intervention and prevention strategies at critical life course periods and disease progression turning points. Initiatives to enable researchers greater access to relevant data sources, such as the HDR UK initiative to harmonise datasets for multimorbidity research, is crucial and should become more generalised in order to gain the insight on multimorbidity processes required to feed into prevention and policy makers strategies at a global scale.

**Acknowledgements** We would like to thank Iris Ho and Bruce Guthrie for providing additional references that our systematic search might have missed.

**Contributors** GC and CTM participated in the screening, data extraction and analysis and helped draft the manuscript. FS and JKFB helped draft the manuscript. KK acquired the funding, conceptualised the study, participated in the screening, data extraction and analysis, helped draft the manuscript and acts as the guarantor for the study. All authors approved the final version for submission.

**Funding** This work was supported by the Academy of Medical Sciences, the Wellcome Trust, the Government Department of Business, Energy and Industrial Strategy, the British Heart Foundation Diabetes UK, and the Global Challenges Research Fund (Grant number SBF004\1093 awarded to KK).

**Disclaimer** The funders had no input to the design or execution of the study.

**Competing interests** None declared.

**Patient consent for publication** Not applicable.

**Provenance and peer review** Not commissioned; externally peer reviewed.

**ORCID iDs**
Genevieve Cezard http://orcid.org/0000-0002-3011-7416
Calum Thomas McHale http://orcid.org/0000-0002-9274-7261
Katherine Keenan http://orcid.org/0000-0002-9670-1607

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
