## [Reviewer comments · BMJ Open]

ARTICLE DETAILS

TITLE (PROVISIONAL)	Studying trajectories of multimorbidity: a systematic scoping review of longitudinal approaches and evidence
AUTHORS	Cezard, Genevieve; McHale, C; Sullivan, Frank; Bowles, Juliana; Keenan, Katherine

VERSION 1 – REVIEW

REVIEWER	Caballero, Francisco Félix Universidad Autonoma de Madrid
REVIEW RETURNED	16-Feb-2021

GENERAL COMMENTS	I have just read the manuscript entitled “Studying trajectories of multimorbidity: a systematic scoping review of longitudinal approaches and evidence”. This revision was focused on longitudinal studies on multimorbidity which were published in the last decade. It is a well written manuscript, and I have only some comments: - Why PubMed was not used for the selection of articles?- Why the article was only focused on recent publications? There were publications on this topic previous to 2010?- Could be these results meta-analysed to strengthen the conclusions?
--

REVIEWER	Kuupiel , Desmond University of KwaZulu-Natal College of Health Sciences, Public Health Medicine
REVIEW RETURNED	27-Feb-2021

GENERAL COMMENTS	Reviewer: Desmond Kuupiel (Ph.D.) Thank you for the opportunity to review this study “Studying trajectories of multimorbidity: a systematic scoping review of longitudinal approaches and evidence.”. This study is relevant with the results a well presented and adequately discussed but some relevant information is missing in the introduction and methods sections. These must be addressed before its acceptance and publication. My comments are outlined below: Introduction 1. What is known about the global burden of multimorbidity?2. At what rate is the global prevalence of multimorbidity projected to increase through the 21st century?3. A statement is made “... United Kingdom, the prevalence of ‘complex multimorbidity’- defined as four or more co-occurring chronic conditions- is projected to double within the next 15 years”
--

I think it is relevant to mention the current prevalence (What is known) in the UK.

4. Aside from systematic reviews, several published scoping reviews focused on multimorbidity. There, it is important to look at these earlier scoping reviews and justify this study based on their limitations. Scoping reviews by their nature permit the inclusion of all study designs. So some prior studies may have reported on longitudinal study designs employed to study multimorbidity.

5. What is this study's main question?

6. "We systematically review the literature via a scoping review approach rather than a systematic review approach [15]. In reporting, we follow the recently developed Preferred Reporting Items for Systematic Reviews and Meta-Analyses for scoping reviews (PRISMA-ScR) [16]" I suggest you add this to the methods section. Nonetheless, why did you choose a scoping review approach instead of a systematic review approach?

Methods

1. There are known methodological frameworks/guidance for the conduct of scoping reviews such as the Arksey and O'Malley framework, Joana Briggs Institute guideline, etc. Which of them did you employ for this review?

2. How was the eligibility of this study's main question determined e.g PCC, PICO/PECO? Please clarify this or provide a table illustrating it.

3. There is no mention of a study protocol preceding the review although I have sighted the preprint. Why?

4.

Search strategy

1. The search strategy did not consider Medical Subject Headings (MeSH) terms. Why?

2. What tool was used to compile the studies?

3. Only peer-reviewed papers were sourced. Why? Nonetheless, the exclusion criterion was silent on unpublished literature. I think it is essential to included relevant unpublished studies.

Screening and study selection

1. Screening and selection of studies were independently conducted by three reviewers at the full text stage but the same was not done at the title and abstract screening phases. So, how did you do to reduce selection bias at the title, and abstract screening phases?

2. How did you ensure the rigor of the screening tools?

3. Since discrepancies were resolved via discussion, it will be relevant to calculate the inter-rater agreement following the full text screening phase.

Data extraction

1. How was the data extracted (deductive or inductive or both approaches)?

2. "...examined bias and limitations to identify knowledge gaps and the strengths and weaknesses of methodological approaches" Please how did examine the bias and the methodological strengths and weaknesses in this review? It is important to include this information.

3. How was the data analysed?

Results

Well presented

	Discussion Well discussed but very few limitations discussed I hope this helps. Thank you
--	--

VERSION 1 – AUTHOR RESPONSE

Reviewer: 1

Dr. Francisco Félix Caballero, Universidad Autonoma de Madrid

Comments to the Author:

I have just read the manuscript entitled “Studying trajectories of multimorbidity: a systematic scoping review of longitudinal approaches and evidence”. This revision was focused on longitudinal studies on multimorbidity which were published in the last decade. It is a well written manuscript, and I have only some comments:

We thank the reviewer for their positive comments.

- Why PubMed was not used for the selection of articles?

We have used Medline which includes approximately 98% of PubMed. Note we have also used three other databases including large interdisciplinary databases such as Web of Science and Scopus, all of which overlap to some extent.

- Why the article was only focused on recent publications? There were publications on this topic previous to 2010?

“Our search had no restrictions on date of publication.” is now specified at the end of the eligibility criteria paragraph in the Methods section. Our search resulted in a selection of more recent publications because the field of multimorbidity is a relatively recent topic of research (mostly published in the last two decades) and the analysis of longitudinal multimorbidity data is even more recent. Therefore, the dates of publication of our selected papers reflect the contemporary nature of this field of research. Our search found no relevant publications published prior to 2010.

- Could be these results meta-analysed to strengthen the conclusions?

No, the methods and the type of estimates produced (when some are produced) are too heterogenous for a meta-analysis exercise. There are also a wide range of multimorbidity research topics with longitudinal approaches. This review is a scoping review which aims to scope the methods and evidence generated by this nascent longitudinal multimorbidity research using a narrative synthesis.

We have added more details on this in the last paragraph of the introduction as follows:

“We used a scoping review approach to systematically review the emerging body of literature of longitudinal studies that have investigated multimorbidity trajectories. Based on a narrative synthesis focused on commonalities and differences, this review provides a methodological summary and a comprehensive synthesis of the evidence on factors affecting multimorbidity pathways.”

In the first paragraph of the discussion, we had explained the following:

“The scoping review process meant that we summarized a wide variety of evidence, and therefore it was not possible to perform a meta-analysis or use a standardized critical appraisal tool. Nevertheless, we provide a narrative-style critical summary of the selected articles.”

Reviewer: 2

Dr. Desmond Kuupiel, University of KwaZulu-Natal College of Health Sciences, Research for Sustainable Development Consult

Comments to the Author:

Reviewer: Desmond Kuupiel (Ph.D.)

Thank you for the opportunity to review this study “Studying trajectories of multimorbidity: a systematic scoping review of longitudinal approaches and evidence.” This study is relevant with the results a well presented and adequately discussed but some relevant information is missing in the introduction and methods sections. These must be addressed before its acceptance and publication.

We thank the reviewer for their positive comments on the results and discussion sections. We trust that we have adequately addressed the points raised by the reviewer regarding the introduction and methods sections in our point-by-point response below.

My comments are outlined below:

Introduction

1. What is known about the global burden of multimorbidity?

We added one sentence in the first paragraph of the introduction section in order to provide an idea of the prevalence of multimorbidity globally and as follows:

“It is estimated that 50 million people suffer from multimorbidity in the European Union, and about one in three globally have multiple conditions [3,4].”

Note that as the definition and measurement of multimorbidity vary across countries, the prevalence of multimorbidity has been found to vary from 3.5% to 100% in a systematic review of multimorbidity studies by Xu, Mishra & Jones published in 2017 (cited in our manuscript).

Furthermore, the Academy of Medical Sciences explains: “It [multimorbidity] is reported to affect anywhere between 13-95% of patients globally, a range so wide that it indicates just how little is known about this global burden.” ([Global burden of multiple serious illnesses must be urgently addressed | The Academy of Medical Sciences \(acmedsci.ac.uk\)](https://www.acmedsci.ac.uk/global-burden-of-multiple-serious-illnesses-must-be-urgently-addressed))

In relation to burden on individual and society/health care, we have already mentioned in the first paragraph of the introduction section the following:

“The implications of this for individuals and societies are stark: multimorbidity is predictive of poorer quality of life [6], greater functional decline [7], and increased mortality [8]. Management and treatment of multimorbidity also places a considerable economic and logistical burden on health services [9], which are not adapted to deal with multimorbidity, being typically organised around the single disease model.”

2. At what rate is the global prevalence of multimorbidity projected to increase through the 21st century?

As multimorbidity prevalence is already difficult to estimate accurately globally, there is no estimation on the rate of increase in multimorbidity prevalence at the global level. It is nevertheless expected to increase as the population continue to grow and age.

One paper by Kingston et al. has made projections based on English data and is cited in this review. Another paper by Laires & Perelman estimated the prevalence of multimorbidity in a Southern Europe population and projected a growth of 13% up to 2050. We would nevertheless prefer to refrain from providing an actual growth rate until further research confirms the scale of what should be expected.

3. A statement is made “...United Kingdom, the prevalence of ‘complex multimorbidity’- defined as four or more co-occurring chronic conditions- is projected to double within the next 15 years” I think it is relevant to mention the current prevalence (What is known) in the UK.

We have now changed this sentence and added both current and projected prevalences as follows:

“For example, the prevalence of ‘complex multimorbidity’- defined as four or more co-occurring chronic conditions - has been projected to increase from about 10% in 2015 to 17% in 2035 in England [5]”.

4. Aside from systematic reviews, several published scoping reviews focused on multimorbidity. There, it is important to look at these earlier scoping reviews and justify this study based on their limitations. Scoping reviews by their nature permit the inclusion of all study designs. So some prior studies may have reported on longitudinal study designs employed to study multimorbidity.

We thoroughly searched for any multimorbidity reviews (both systematic and scoping reviews) prior to beginning our review, which might cover the range of longitudinal methodologies used in multimorbidity studies, but we did not find any. Some of the recent multimorbidity reviews we found are cited in this manuscript such as the systematic review of systematic reviews by Johnson et al. published in 2019 or the systematic review by Stirland et al. published in 2020.

We want to emphasize that this review is filling a clear gap in the world of multimorbidity research with the potential to help others in the field in terms of choice of longitudinal methods and gaps in multimorbidity longitudinal research (see example below of feedback from a London researcher). In the second paragraph of our introduction, we explained that:

“To date there are more than 70 published systematic reviews about multimorbidity, covering definitions to interventions (for example[2, 13]), and none of these focuses on reviewing longitudinal studies.”

Following this, we have now explicitly stated the following:

“This study was designed to fill this clear research gap.”

The preprint version of our review has been viewed and read by multimorbidity researchers, some of whom contacted us. Feedback emphasized the usefulness of such a scoping review in a world where longitudinal multimorbidity research is increasing. One researcher in London wrote (personal communication in May 2021): “Just wanted to let you know I think it is a super paper and I’ve found it immediately helpful - so glad you wrote it. [...] (we are evaluating MM data in South London with 15yr data”.

We strongly believe that this review matters to guide current researchers in the field of longitudinal multimorbidity research in addition to filling a gap not covered by the many multimorbidity systematic and scoping reviews that have been published.

5. What is this study’s main question?

The study aims to scope the longitudinal methods used in researching multimorbidity and disease trajectories overtime as well as the evidence generated from this type of longitudinal approaches. This aim is reflected in our title “Studying trajectories of multimorbidity: a systematic scoping review of longitudinal approaches and evidence.”

We have also stated our research questions at the end of the introduction as follows:

- 1. What type and range of longitudinal methods are used to analyse multimorbidity over time within individuals?*
- 2. What are the risk/protective factors identified to be associated with individual multimorbidity trajectories?*

We have now added a sentence in the introduction prior to our research questions to clarify our general aim as follows:

“Therefore, this paper aims to gain an overview of the longitudinal approaches used in multimorbidity research, to better understand what evidence is generated from these approaches, and to identify the associated gaps.”

6. “We systematically review the literature via a scoping review approach rather than a systematic review approach [15]. In reporting, we follow the recently developed Preferred Reporting Items for Systematic Reviews and Meta-Analyses for scoping reviews (PRISMA-ScR) [16]” I suggest you add this to the methods section. Nonetheless, why did you choose a scoping review approach instead of a systematic review approach?

Thank you for your recommendation. This section has now been moved to the Methods section.

Reference 17 is a paper by Munn et al. published in 2018 which explains when a systematic review or a scoping review is most appropriate. In brief, scoping reviews are conducted to summarise a body of literature and identify knowledge gaps, whereas systematic reviews address more precise questions, such as feasibility or effectiveness of a certain treatment. Our review aimed to identify and scope the extent of the multimorbidity trajectory literature and summarise their approaches and findings. As such, a scoping review approach was most appropriate for us. However, note that our scoping review takes a systematic approach and we followed PRISMA-ScR in reporting which has been developed specifically for scoping reviews.

We have developed this paragraph at the start of the methods section as follows:

“We review the literature on longitudinal multimorbidity studies via a scoping review approach rather than using a systematic review or meta-analytic approach [16]. Scoping reviews are adopted when the purpose of the review is to scope a nascent body of literature and appraise gaps [17,18]. In reporting, we follow the recently developed Preferred Reporting Items for Systematic Reviews and Meta-Analyses for scoping reviews (PRISMA-ScR) [19] (Appendix A).”

Methods

1. There are known methodological frameworks/guidance for the conduct of scoping reviews such as the Arksey and O’Malley framework, Joana Briggs Institute guideline, etc. Which of them did you employ for this review?

Munn and al. (2018) and Peters et al. (2015) were cited to justify the choice of a scoping review as the most appropriate for our aim. These citations are listed as resources on the Joana Briggs Institute website in relation to scoping reviews. We have also followed a double data extraction process advised by Busceni et al. (2006) as well as the adapted PRISMA-ScR checklist for scoping reviews by Tricco et al. (2018).

2. What tool was used to compile the studies?

We used Endnote to store and classify all the references into categories of reason for exclusion or inclusion at each stage of the screening process. We now specify this in the following sentence of the “Screening and study selection” part of the Methods section:

“After de-duplication, articles were screened for eligibility by title, abstract and finally full text (work shared between GC, CM, and KK) using Endnote and pre-defined groups for exclusion reasons and inclusion.”

3. Only peer-reviewed papers were sourced. Why? Nonetheless, the exclusion criterion was silent on unpublished literature. I think it is essential to included relevant unpublished studies.

We made the decision to include solely peer-reviewed published manuscripts. This was a scoping review and, although we aimed to conduct a thorough and systematic search process, we did not aim to conduct an exhaustive search of the literature. We did not search for grey literature because of concerns about

resources and search efficiency as well as the replicability and accuracy of the search strategy out with indexed databases.

Screening and study selection

1. Screening and selection of studies were independently conducted by three reviewers at the full text stage but the same was not done at the title and abstract screening phases. So, how did you do to reduce selection bias at the title, and abstract screening phases?

After the search and the deduplication process, over 6000 titles were split evenly between three reviewers and were independently title screened. Categories of exclusion were pre-defined and were created in Endnote using groups. A group for “potential inclusion” was also created. Each researcher could then assign each paper into the right group. If in doubt, a paper would be classified into the “potential inclusion” group and kept for the next stage and another reviewer would be assigned that list of papers to review at a subsequent stage. This process was repeated for abstract screening.

At the abstract screening stage, a second reviewer double-screened the abstract kept for inclusion. Following your comment, we have further improved this process by double screening the abstracts that were not kept and we have updated the text as follows:

“At abstract and full text stages, a double screening process was used to minimise evidence selection bias [21], meaning two co-authors blindly and independently reviewed the study for inclusion.”

Due to this additional process, we have now included an abstract for which exclusion was not clear cut, a paper by Freisling et al. published in 2020 which was initially missed at the abstract screening stage. The figure, tables, and corresponding results have been updated accordingly.

How did you ensure the rigor of the screening tools?

As written above, we used three independent reviewers at all stages of the process, double screened at abstract and full text stage, followed pre-defined criteria and standardised templates. Any papers we were undecided about were discussed as a group of three reviewers. We ensured rigor through using a pre-determined, principled systematic process which we all followed.

3. Since discrepancies were resolved via discussion, it will be relevant to calculate the inter-rater agreement following the full text screening phase.

Uncertainties about inclusion/exclusion were discussed and this ensured 100% agreement was reached.

Data extraction

1. How was the data extracted (deductive or inductive or both approaches)?

We used both approaches.

The data was extracted using a spreadsheet with pre-defined categories (described in our Data extraction and synthesis section of the Methods) using a deductive approach. The extracted data further fed into sub-tables with topics such as study and sample characteristics, data types and sources, multimorbidity measurement, methods including outcomes and risk factors but aims, main findings, and limitations had also been extracted.

From the sub-tables (some of which are provided in the manuscript), we didn't prespecify which type of methods would be found. Therefore, the groups of longitudinal approaches described in our results section were determined from the data and in some way, followed an inductive approach.

2. "...examined bias and limitations to identify knowledge gaps and the strengths and weaknesses of methodological approaches" Please how did examine the bias and the methodological strengths and weaknesses in this review? It is important to include this information.

Our review is a scoping review of longitudinal studies of multimorbidity focusing on the range of methods that have been used so far. Our aim was to scope rather than to quality appraise studies. However, limitations of studies and methodological approaches were extracted from each paper following key aspects as explained in the "Data extraction and synthesis" part of the Methods section as follows:

"Finally, we extracted the key findings, and limitations reported in each study in relation to generalisability, accuracy, comprehensiveness, methodology, and interpretation."

Some of these extracted limitations such as sample size, generalisability issues, source of bias and the heterogeneous measurement of multimorbidity are mentioned in the discussion section (fourth paragraph). Outside of the data extraction process, we also discussed the advantages and disadvantages of each specific group of methods and what it means to choose one type of method rather than another for analysing multimorbidity (fifth and sixth paragraphs of our discussion section).

3. How was the data analysed?

We used a narrative synthesis based on themes sub-tables created from the master data extraction table. The narrative syntheses extracted data based on commonalities and differences in themes sub-tables. This is now summarised at the end of the Introduction as follows:

"Based on a narrative synthesis focused on commonalities and differences, this review provides a methodological summary and a comprehensive synthesis of the evidence on factors affecting multimorbidity pathways."

This is also specified at the end of the "Data extraction and synthesis" part of the methods section as follows:

"To develop the narrative synthesis, we analysed and summarised the patterns in the extracted data, investigated the similarities and differences between studies, and examined bias and limitations to identify knowledge gaps and the strengths and weaknesses of methodological approaches."

Results

Well presented

Discussion

Well discussed but very few limitations discussed I hope this helps.

We have added a second paragraph in the discussion section which explains the methodological limitations of our study and the logic behind some methodological choices for this review as follows:

"A limitation of narrative reviews is that they might select evidence to support a particular stance and do not necessarily take enough steps to eliminate selection bias. However, we selected a comprehensive set of items to extract before starting the review and we engaged in double screening and extraction. Therefore, our methodological approach should limit selection and extraction bias. In addition, our review didn't engage in a critical appraisal of the quality of the selected studies. However, the aim of the review was to scope the published evidence on longitudinal approaches to multimorbidity. Indeed, when the aim of a scoping review is to provide an overview of evidence, methodological limitations and risk of bias of the evidence are not necessarily relevant and generally not performed [18]."